# A reduction in self-reported confidence accompanies the recall of memories distorted by prototypes
Casper Kerrén[1], Yiming Zhao[2] & Benjamin J. Griffiths [2] ✉

When we recall a past event, we reconstruct the event based on a combination of episodic details and semantic knowledge (e.g., prototypes). Though prototypes can impair the veracity of recall, it remains unclear whether we are metacognitively aware of the distortions they introduce. To address this, we conducted six experiments in which participants learned object-colour/object-location pairs and subsequently recalled the colour/location when cued with the object. Leveraging unsupervised machine learning algorithms, we extracted participant-specific prototypes and embedded responses in two-dimensional space to quantify prototype-based distortions in individual memory traces. Our findings reveal robust and conceptually replicable evidence to suggest that prototype-based distortion is accompanied by a reduction in self-reported confidence - an implicit measure of metacognitive awareness. Critically, we find evidence to suggest that it is prototype-based distortion of a memory trace that undermines confidence, rather than a lack of confidence biasing reconstruction towards the use of prototypes. Collectively, these findings suggest that we possess metacognitive awareness of distortions embedded in our memories.

We remember the past by entwining episodic details with semantics, schemas, and prototypes[1–6]. For instance, the recollection of a morning commute may be built from prototypes of the usual route and means of transportation incorporated with details unique to that trip (e.g., roadworks exacerbating traffic). While the veracity of a memory hinges upon the extent to which reconstruction relies on episodic (over non-episodic) details[4,5,7], it remains unclear whether we possess metacognitive awareness of distortions introduced by non-episodic details. Here, we set out to address this question.

Memories are susceptible to influence from prototypes. For example, when participants are tasked with encoding dots presented within a circle, subsequent recollection of individual dots will be biased by the centre mass of other nearby dots[5]. This prototypical bias has been observed for a range of stimuli, including spatial locations[3,5,7–10], objects[4,6,11,12], colours[13–18], faces[19–21] and words[22–24]. Importantly, these influences have real-world consequences such as in eyewitness testimony[25–28]: misidentification may occur when recalling an individual face that aligns with a "criminal" stereotype[29], which may result in wrongful convictions[30].

But to what extent are we metacognitively aware of these prototype-based distortions? While an explicit question about prototypes may in and of itself induce metacognitive awareness, it is possible to probe metacognitive awareness implicitly using confidence ratings. Indeed, a study of particular relevance has shown that when we actively recall memories that

are distorted by prototypes, confidence declines[31]. However, it remains to be seen whether this lack of confidence (i) reflects metacognitive awareness of prototype-based distortions embedded in a memory trace, or (ii) drives us to favour prototypes over episodic details during reconstruction (e.g.[5,32]). Delineating these hypotheses would offer key insights into how and when prototype-based distortions influence memory, and whether these distortions can be disentangled from a memory trace.

We conducted six experiments using a cued recall precision memory task, investigating how subjective confidence ratings relate to the prototype-based distortion (termed "prototypicality") of long-term memory traces. Participants learned object-colour or object-location associations and later recalled the colour/location using the object as a cue. By using a k-means clustering algorithm to derive participant-specific prototypes and then embedding memory responses in two-dimensional space to measure prototypicality, we reveal a robust negative correlation between prototypicality and confidence. We then iterate through several experimental designs to deduce why this relationship exists.

## Methods

### Pre-registration

The hypotheses of Experiment 1 were pre-registered on 17th July 2023 (https://osf.io/v925d). As the pre-registered clustering measure resulted in a

[1]Max Planck Institute for Human Cognitive and Brain Sciences, Leipzig, Germany. [2]Centre for Human Brain Health, University of Birmingham, Birmingham, UK. ✉e-mail: b.griffiths.1@bham.ac.uk

substantial degree of participant attrition (~42%) and did not account for trial-by-trial variations in the distance between targets and prototypes, we developed a new measure ("prototypicality"), for which all following analyses are based upon. For transparency, the original pre-registration has been presented in Supplementary Note 1 and Supplementary Fig. 1 alongside any deviations from the protocol and the results of the pre-registered analyses. These results largely match the analyses we report below. We did not pre-register the subsequent experiments.

### Participants

A total of 218 participants were recruited (~71.3% female [self-reported]; mean age = ~24.0 years [s.d.: ~5.8 years]; no information on race/ethnicity was collected; for breakdown by experiment, see Table 1). The sample size of Experiment 1 was determined based on sample sizes of similar studies[31]. The remaining experiments used sample sizes that ensured >95% power based on the data obtained in Experiment 1. For Experiments 1 and 3, participants were recruited using Prolific (www.prolific.com) and received financial reimbursement for their time. Participants reported being fluent in English, aged between 18 and 35, and residing in the United Kingdom. For Experiments 2, 4, 5, and 6, participants were recruited using the University of Birmingham Research Participation Scheme and received course credit for their participation. All participants reported being aged between 18 and 35, with fluency in English and residence in the United Kingdom being assumed given they had enroled in the university psychology undergraduate course. Data collection began in July 2023 and ended around October 2023. There were momentary gaps in collection between the experiments, but no gaps within an experiment. All participants provided informed consent before taking part. Ethical approval was granted by the Research Ethics Committee at the University of Birmingham.

### General study design

All experiments followed the same overarching design (see Fig. 1a) and were delivered online via the Gorilla Experiment Builder platform

### Table 1 | Participant samples by experiment

| Experiment | Recruitment platform | Sample size | Mean age | % Female |
|---|---|---|---|---|
| 1 | Prolific | 45 | 28.6 | 51.1 |
| 2 | University Research Participation Scheme | 34 | Not available | |
| 3 | Prolific | 29 | 27.0 | 51.7 |
| 4 | University Research Participation Scheme | 37 | 20.3 | 94.6 |
| 5 | University Research Participation Scheme | 41 | Not available | |
| 6 | University Research Participation Scheme | 33 | 19.2 | 90.6 |

Experimenter error meant that demographics for Experiment 2 and 4 were not collected. However, given that these samples were drawn from the same pool as Experiment 3 and 5 during the same period (autumn/winter 2023), it is reasonable to assume a high degree of similarity in demographics between these samples.

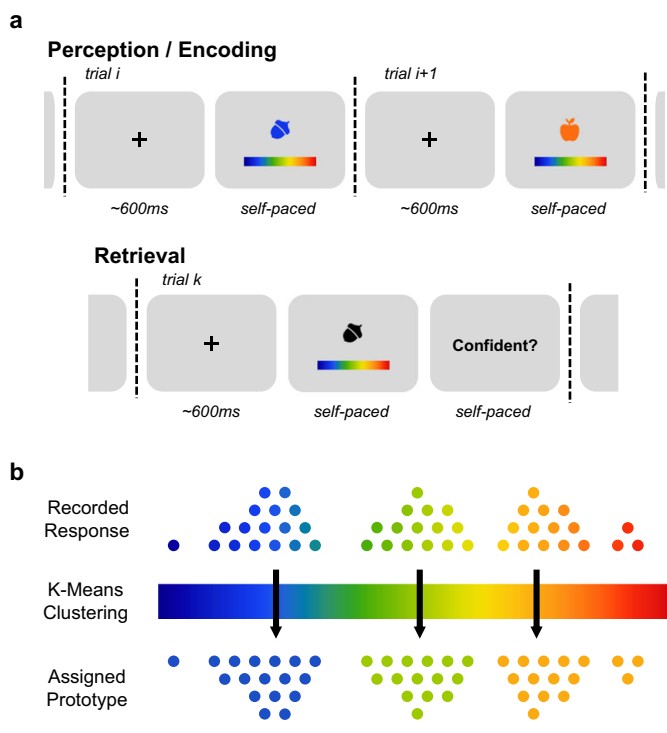

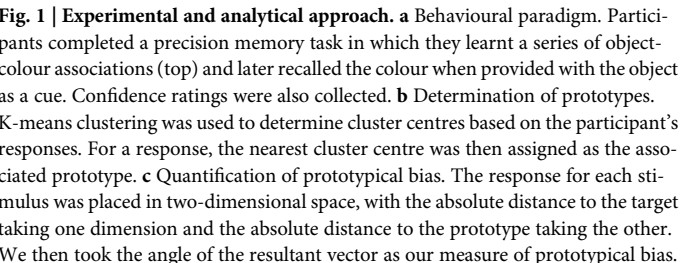

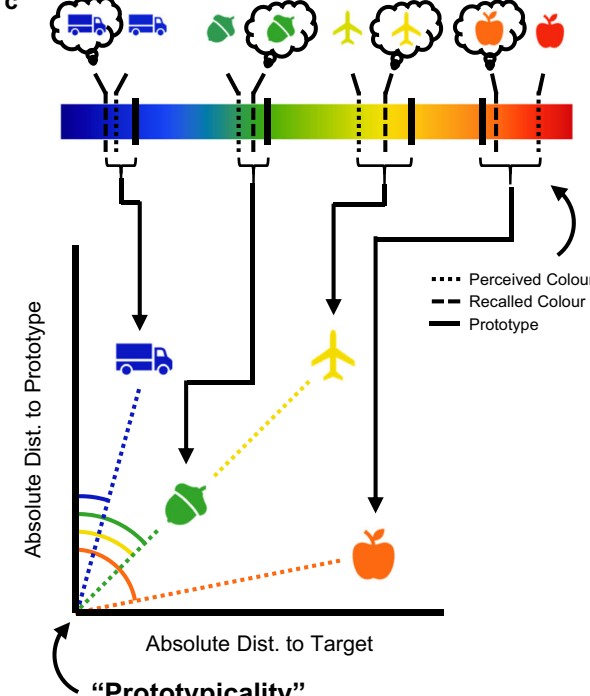

**Fig. 1 | Experimental and analytical approach. a** Behavioural paradigm. Participants completed a precision memory task in which they learnt a series of object-colour associations (top) and later recalled the colour when provided with the object as a cue. Confidence ratings were also collected. **b** Determination of prototypes. K-means clustering was used to determine cluster centres based on the participant's responses. For a response, the nearest cluster centre was then assigned as the associated prototype. **c** Quantification of prototypical bias. The response for each stimulus was placed in two-dimensional space, with the absolute distance to the target taking one dimension and the absolute distance to the prototype taking the other. We then took the angle of the resultant vector as our measure of prototypical bias.

When a response is close to the prototype and far from the target, prototypical bias is high (see orange-coloured apple). Importantly, two items can have differing distances to the prototype but the same prototypical bias if the distance to the target varies between the two stimuli (see blue-coloured acorn and green-coloured aeroplane), demonstrating how this measure avoids biases introduced by trial-by-trial variability in the distance between the target colour and the prototype. In instances when a response is repulsed by a prototype (see blue truck), prototypical bias is very low. In instances when a response overshoots its prototype (not visualised), prototypical bias is very high. Excluding repulsions and overshoots has no impact on the central results (see Supplementary Table 1).

**Table 2 | Study design variations**

| Experiment | Modality | Sampling distribution | Training | Other notes |
|---|---|---|---|---|
| 1 | Colour | Ten clusters | One training block | N/A |
| 2 | Colour | Uniform | Self-paced task walkthrough | On half of the retrieval trials, the colour bar was reversed. |
| 3 | 1D location | Ten clusters | One training block | Block size varied from 6 trials to 10 trials, but the block size manipulation was not analysed here. |
| 4 | Colour | Uniform | Self-paced task walkthrough | On half of the retrieval trials, the confidence rating was presented before the colour decision. |
| 5 | Colour | Two broad clusters (-10% to +10% of whole colour bar, around cluster centre), two narrow clusters (−3% to +3% of whole colour bar, around cluster centre) | Self-paced task walkthrough | Eight trials were used per block to ensure equal sampling from the four kernels within a block. The number of blocks increased to 15 to compensate. |
| 6 | Colour | Uniform | Self-paced task walkthrough | Confidence was measured at encoding. |

(www.gorilla.sc[33]). The following study design describes Experiment 1; variations in study design between Experiment 1 and the other experiments are listed in Table 2. The task took approximately 30 min to complete; participants who failed to complete an experiment in 1 h were presumed to have engaged in another activity mid-experiment and automatically discarded by Gorilla Experiment Builder.

During encoding, participants were presented with easily recognisable, coloured objects (taken from Microsoft Office "Icons"; inspired by[34]) and asked to (i) encode the object-colour association and (ii) report the perceived colour. Colours were drawn from a colour bar that was not presumed to be perceptually uniform as, given individual variability in colour perception[35], we felt no colour bar could be considered truly uniform for all individuals. Instead, we used the perceptual judgement as a reference point for what could be considered the "target" during memory retrieval. This approach ensured that perceptual biases did not introduce a source of uncontrolled variability between individual object-colour associations. The object colours were drawn from a distribution with ten latent clusters. Participants were not informed about this distribution. After being exposed to ten of these pairs, participants completed ten maths sums that served as a distractor task. Each sum took the form of subtracting a single digit number from a three-digit number. The answer and a lure were presented on screen and participants made a forced-choice; if they selected incorrectly, they were re-presented with the question. During retrieval, participants were presented with the object as a cue and asked to recall the colour by clicking on the appropriate position on the colour bar. Participants were presented with three confidence ratings ("Sure", "Unsure", and "Guess") and were forced to choose between the three. In all experiments, participants provided a confidence rating once per stimulus. In most experiments, this was during memory retrieval; the exception was Experiment 6, where a prospective confidence rating was collected during memory encoding (see Table 2).

Participants completed 13 blocks of the task, each consisting of 10 object-colour pairs. The first block acted as a training block and was not analysed, leaving a total of 12 blocks (120 pairs) for subsequent analysis.

### Analysis

All analysis was conducted using custom scripts written in Python 3. Statistical results were cross-checked in JASP 0.18.1.0. Bayes Factor was computed using JASP 0.18.1.0. Confidence intervals for Cohen's d and partial eta-squared effect sizes were computed using an online tool: https://effect-size-calculator.herokuapp.com/.

The principal analyses focused on the relationship between the prototype-based distortion ("prototypicality") of a memory trace and subjective ratings of confidence. Prototypicality can be thought of as the extent to which an episodic memory shifts from veridical representation of a stimulus towards a prototypical version. If there is little change in the representation between perception and retrieval, prototypicality is said to be low. If the representation shifts drastically towards the prototype between perception and retrieval, prototypicality is said to be high.

Prototypes were derived from the data for each participant individually using cross-validated (here, *leave-one-out*) k-means clustering. For every response of every participant, we pooled all responses, excluding the response of interest, and derived $k$ clusters, with $k$ iteratively taking all integer values between 2 and 10 (inclusively). For each value of $k$, the k-means clustering algorithm was run 300 times, with the cluster centroids defined as the series of 10 runs of these runs which produced optimal inertia. Optimal inertia refers to the iteration which produces the smallest sum of all squared distances between data points within a cluster and the cluster centroid. The optimal number of clusters was then defined as the $k$ with the largest mean silhouette score. The silhouette score describes how well one cluster is separated from others; a high silhouette score indicates a good separation between clusters. For full details, see the *scikit-learn* API. This approach resulted in $k$ clusters for each participant, with the centroid of each cluster representing the putative prototype colour/location (see Fig. 1b). A summary of descriptive statistics regarding these clusters is presented in Table 3. Broadly speaking, each experimental paradigm resulted in a similar

## Table 3 | K-means cluster descriptive statistics

| Experiment | N. latent clusters | N. clusters | |
|---|---|---|---|
| | | Mean (SEM) | Median |
| 1 | 10 | 4.30 (0.34) | 3 |
| 2 | 0 | 3.98 (0.40) | 3 |
| 3 | 0 | 2.07 (0.05) | 2 |
| 4 | 4 | 4.67 (0.37) | 4 |
| 5 | 0 | 4.10 (0.37) | 3 |
| 6 | 10 | 5.34 (0.37) | 4 |

mean number of clusters being formed, with underlying latent distributions having little impact on the mean number of clusters formed.

Prototypicality was computed by placing the response of interest in two-dimensional space (see Fig. 1c). The first dimension reflected the distance to the target. For the perceptual response, the target reflected the presented colour; for the retrieval response, the target reflected the colour perceived at encoding. We referenced the retrieval data to the perceptual response to subtract perceptual distortions out of the retrieval effects, ensuring retrieval effects reflect memory-based distortions. The second dimension reflected the distance to the nearest prototype (i.e., the k-means cluster centres). We took the angle of this vector as our measure of prototypicality. Note that this approach accounts for variability in the distance between the prototype and each stimulus colour that may bias measures looking solely at distance to the prototype.

In all but Experiment 3 and 6, confidence was collected on a three-point scale. However, responses for positions lower on the confidence scale were sparse (percentage of responses marked as "Sure": 60.2%; "Unsure": 23.8%; "Guess": 16.0%), with many participants only making use of the "Sure" response and one of the "Unsure" or "Guess" options. Therefore, we collapsed across the two lower confidence options to provide a binary measure of confidence: "Sure" (60.2% of responses) and "Not Sure" (49.8% of responses).

We also factored the epoch of the response into our analyses ("perception" or "retrieval"). This allowed us to track how distortion affects change between the perception and recall of a stimulus.

Taking these variables together, we conducted a $2 \times 2$ repeated measures ANOVA, with prototypicality acting as the dependent variable and confidence and epoch acting as the independent variables. Data distribution was assumed to be normal but this was not formally tested. A result was considered significant when the associated $p$-value was less than 0.05.

Experiments 3, 4 and 5 each contained an additional binary variable of interest (Exp. 3: colour bar orientation; Exp. 4: confidence/colour judgement order; Exp. 5: kernel size). For these experiments, we expanded the repeated ANOVA into a three-factor model, adding the experiment-specific variable to the variables outlined in the previous paragraph. Furthermore, as our explicit aim in Experiment 5 was to bias confidence ratings by making colours more prototypical, we asked whether the proportion of "Sure" responses changed as a function of kernel size. To this end, we computed the proportion of "Sure" responses for "broad" and "narrow" kernels separately and then conducted a repeated measures t-test. As above, a result was considered significant when the associated p-value was less than 0.05.

To examine the correlation between perceptual prototypicality and recall prototypicality, we used Pearson's correlation coefficient to correlate the two metrics across trials for each participant individually. We then pooled the correlation coefficients across participants and subjected them to a one-sample t-test. In Experiment 6, we expanded this analysis to include prospective confidence as a covariate. To do so, we conducted a multiple regression analysis for each participant where recall prototypicality acted as the dependent variable and perceptual prototypicality, prospective confidence and a constant were used as predictor variables. We pooled the beta values across participants and subjected them to one-sample t-tests (for each predictor separately). As above, a result was considered significant when the associated p-value was less than 0.05.

Lastly, following a reviewer's suggestion, we explored whether reaction time fluctuates as a function of prototypicality. However, we found no relationship between the two variables (for further details, see Supplementary Fig. 2).

### Reporting summary
Further information on research design is available in the Nature Portfolio Reporting Summary linked to this article.

## Results
### Confidence declines with prototype-based distortion of memory
We first asked whether a correlation exists between self-reported confidence (a proxy for metacognitive awareness[36]) and prototype-driven biases during recall. Here, participants encoded object-colour associations where the presented colours were sampled from a latent, skewed distribution consisting of 10 prototypes, equidistantly positioned across the colour bar. On average, participants reported recalling the target colour with a high degree of confidence (i.e., a "Sure" response) on 57.5% of trials (standard error of the mean [SEM]: 2.8%). Participant responses were accurate during perception/encoding, then declined during retrieval. The decline was driven by a loss in trial-specific colour information (in Fig. 2a, see the shift in the vector towards the x-axis). This implies that we tend to preserve/recall prototypical details over those episode-specific details.

To statistically quantify how prototypicality changes as a function of trial type and confidence, we conducted a $2 \times 2$ repeated measures ANOVA with epoch and confidence acting as independent variables and prototypicality acting as the dependent (see Fig. 2b). This revealed that (i) colour selections were more prototypical when the colour was retrieved from memory relative to when it was presented on screen [$F(1, 44) = 121.69$, $p < 0.001$, $\eta_p^2 = 0.73$, 95% CI = (0.58, 0.81)], (ii) confidence decreased when making more prototypical decisions [$F(1, 44) = 162.69$, $p < 0.001$, $\eta_p^2 = 0.79$, 95% CI = (0.66, 0.85)], and (iii) the correlation between confidence and prototypicality was more pronounced during retrieval [$F(1, 44) = 153.12$, $p < 0.001$, $\eta_p^2 = 0.78$, 95% CI = (0.64, 0.84)]. This suggests that confidence in a memory-based response declines with the prototypicality of that response.

We then set out to conceptually replicate these results in two experiments that decoupled colour information from spatial location. In Experiment 2, we flipped the colour bar on half of the retrieval trials to rule out the possibility that spatial information impacted (and possibly further distorted) recalled representations. In Experiment 3, colour was removed from the stimuli and participants encoded object-location associations, with locations being uniformly sampled in 1-dimensional space.

In Experiment 2, participants recalled the target colour with a high degree of confidence (i.e., a "Sure" response) on 62.9% of trials (SEM: 2.9%). In a three-factor repeated measures ANOVA (see Fig. 2c–f), we replicated the results of Experiment 1, continuing to observe main effects for epoch [$F(1, 33) = 309.11$, $p < 0.001$, $\eta_p^2 = 0.90$, 95% CI = (0.83, 0.93)], confidence [$F(1, 33) = 127.76$, $p < 0.001$, $\eta_p^2 = 0.79$, 95% CI = (0.64, 0.86)], and an interaction between the two [$F(1, 33) = 165.12$, $p < 0.001$, $\eta_p^2 = 0.83$, 95% CI = (0.71, 0.89)]. We found no main effect for the orientation of the colour bar [$F(1, 33) = 0.54$, $p = 0.488$, $\eta_p^2 = 0.02$, 95% CI = (0.00, 0.17)], nor did we observe an interaction between colour bar orientation and the other factors [orientation*epoch: $F(1, 33) = 0.11$, $p = 0.745$, $\eta_p^2 < 0.01$, 95% CI = (0.00, 0.12); orientation*confidence: $F(1, 33) = 0.09$, $p = 0.763$, $\eta_p^2 < 0.01$, 95% CI = (0.00, 0.12); orientation*epoch*confidence: $F(1, 33) < 0.01$, $p = 0.945$, $\eta_p^2 < 0.01$, 95% CI = (0.00, 0.06)]. A Bayesian ANOVA indicated that spatial location did not exert a significant influence over how we recall colour memories (effect of colour bar orientation: $BF_{10} = 0.14$; orientation*epoch: $BF_{10} = 0.13$; orientation*confidence: $BF_{10} = 0.16$; orientation*epoch*confidence: $BF_{10} = 0.05$).

In Experiment 3, participants recalled the target colour with a high degree of confidence (i.e., a "Sure" response) on 62.4% of trials (SEM: 3.3%). They also proved to be more precise for spatial locations than for colours (see Fig. 2g, h), but nonetheless showed a greater loss of trial-specific information relative to prototypical information from perception to retrieval. Inferential statistics

**Fig. 2 | Task performance in Experiments 1–3.**
**a** Plots of task performance visualised in 2-dimensional space, with distance to the target presented on the x-axis and distance to the proto-type (i.e., the data-derived cluster centres of responses) presented on the y-axis (n = 45 partici-pants). For the perceptual data, the target distances are computed using the perceptual response and the true, presented colour, while the prototype distances are computed using the perceived response and its associated prototype. For the retrieval data, the target distances are computed using the recalled response and the equivalent perceptual response (disentangling memory biases from perceptual bias-es; see methods), while the prototype distances are computed using the recalled response and its asso-ciated prototype. Line plots represent the vector for each condition (grey: perception; coloured: retrie-val). The further the line moves away from the y-axis and towards the x-axis, the greater the proto-typicality. Individual dots represent individual par-ticipants. The density plots above and to the right of the main plots represent the distributions for each dimension separately. **b** Boxplots for data from Experiment 1, depicting prototypicality (i.e., the angle of the vectors in panel (**a**)) given epoch (x-axis) and confidence rating (hue; "Sure" in blue; "Not Sure" in red) for Experiment 1. The boxplots display the median and interquartile range, with the whis-kers capturing the range of the data. Individual dots reflect individual participants (n = 45 participants). All data reflects within-participant variation (Loftus & Masson, 1994). **c**, **d** Two-dimensional perfor-mance plots and boxplots from Experiment 2 when the colour bar direction remained unchanged between encoding and retrieval (n = 34 partici-pants). All plot details match panel (**a**/**b**). **e**, **f** Two-dimensional performance plots and boxplots from Experiment 2 when the colour bar direction was reversed between encoding and retrieval (n = 34 participants). All plot details match panel (**a**/**b**). **g**, **h** Two-dimensional performance plots and box-plots from Experiment 3 (n = 29 participants). All plot details match panel (**a**/**b**).

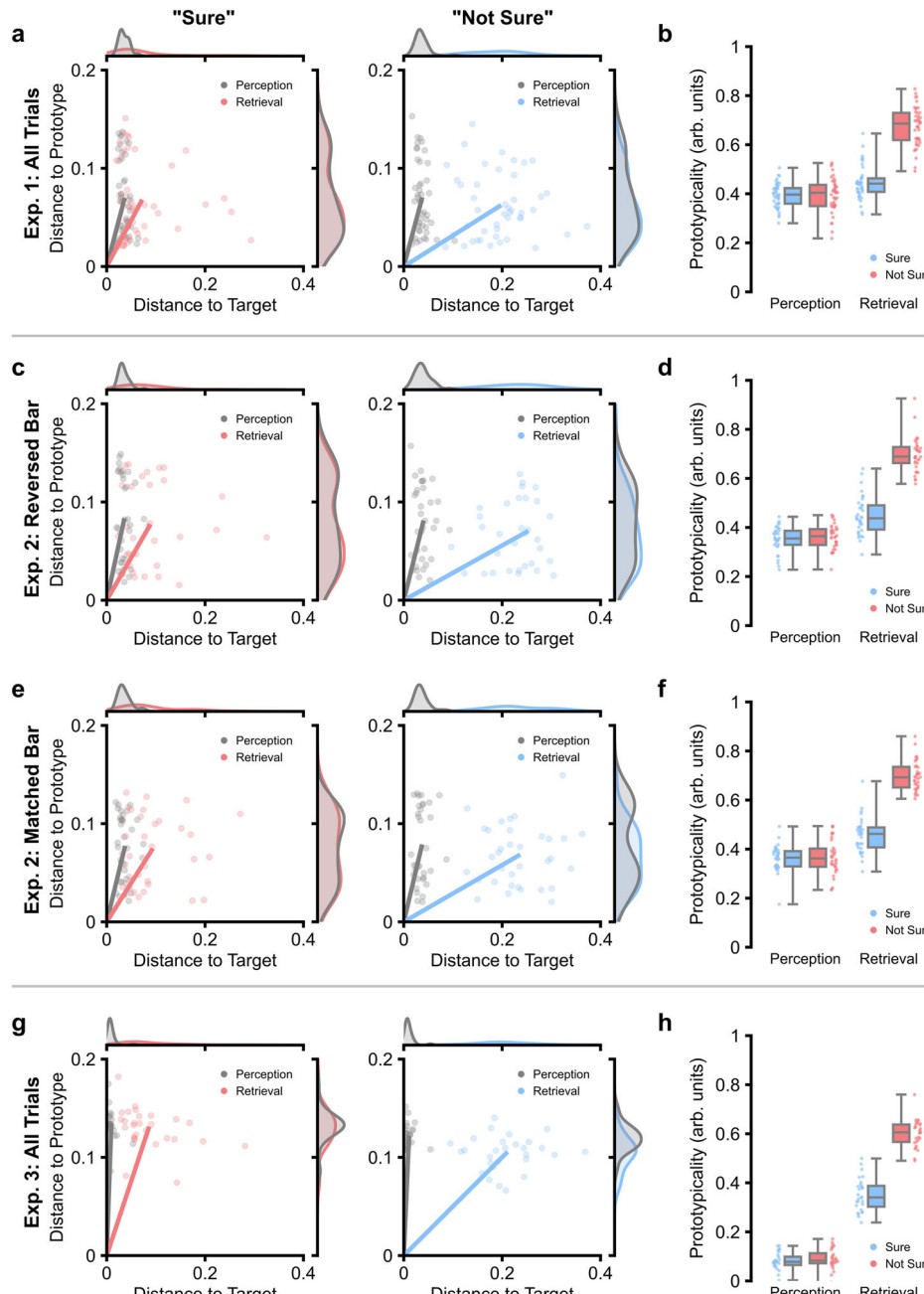

matched those above: there was a main effect of epoch [$F(1, 27) = 908.65$, $p < 0.001$, $\eta_p^2 = 0.97$, 95% CI = (0.94, 0.98)], a main effect of confidence [$F(1, 27) = 158.00$, $p < 0.001$, $\eta_p^2 = 0.85$, 95% CI = (0.72, 0.90)], and an interaction between the two [$F(1, 27) = 148.84$, $p < 0.001$, $\eta_p^2 = 0.85$, 95% CI = (0.71, 0.90)]. These results demonstrate that prototypical memory traces for space bias confidence in much the same way as they do for colour, suggesting this prototype-driven bias occurs across modalities.

Altogether, these results provide replicable and cross-modal evidence to suggest that prototypical responses negatively correlate with confidence, suggesting that we possess metacognitive awareness of prototype-based distortions in memory.

## Prototype-based distortions of memory contribute to the reduction in confidence

Confident in the fact that prototype-based distortions correlate with a reduction in confidence, we then asked why this occurs. We considered three different hypotheses (see Fig. 3a): (i) confidence is derived from the similarity between reconstruction/response and the prototype[37]; (ii) weak memory traces result in a lack of confidence which, in turn, leads partici-pants to make prototypical reconstructions/responses[5]; (iii) confidence is unrelated to reconstruction/response and is instead derived from the pro-totypicality of the memory trace itself.

To investigate the first two of these hypotheses, Experiment 4 altered the order in which participants made their decisions on retrieval trials: on half the trials, participants chose colour first and then reported confidence (as in Experiments 1–2); on the other half of trials, participants reported confidence in their recall, then chose the colour. We reasoned that if confidence is influencing prototypical reconstruction/responding, then priming participants with their confidence rating before colour selection would impact the prototypicality of responses. In contrast, if the selection of a prototypical colour undermines confidence, collecting confidence ratings before colour selection may break the confidence-prototypicality correlation observed in Experiments 1–3.

**Fig. 3 | Task performance in Experiments 4 and 5.**
**a** Visual depiction of three possible explanations for the link between prototypical responding and confidence ratings. Hypothesis 1 proposes that responding in a prototypical manner influences confidence ratings. Hypothesis 2 proposes that a weak memory trace undermines confidence, which in turn leads participants to bias their response towards a prototype. Hypothesis 3 proposes that recall exerts separable effects on responding and confidence ratings. **b** Plots of task performance visualised in 2-dimensional space ($n = 37$ participants). Line plots represent the vector for each condition (grey: perception; coloured: retrieval). The density plots above and to the right of the main plots represent the distributions for each dimension separately. **c** Boxplots depicting prototypicality given epoch (*x*-axis) and confidence rating (hue). The boxplots display the median and interquartile range, with the whiskers capturing the range of the data. Individual dots reflect individual participants ($n = 37$ participants). All data reflects within-participant variation (Loftus & Masson, 1994).
**d, e** Two-dimensional performance plots and boxplots from Experiment 4 when participants selected the recalled colour and then reported confidence ($n = 37$ participants). All plot details match those of panels (**b**) and (**c**). **f, g** Two-dimensional performance plots and boxplots from Experiment 5 for stimuli which were drawn from a broad distribution of colours ($n = 41$ participants). All plot details match those of panels (**b**) and (**c**). **h, i** Two-dimensional performance plots and boxplots from Experiment 5 for stimuli which were drawn from a narrow distribution of colours ($n = 41$ participants). All plot details match those of panels (**b**) and (**c**).

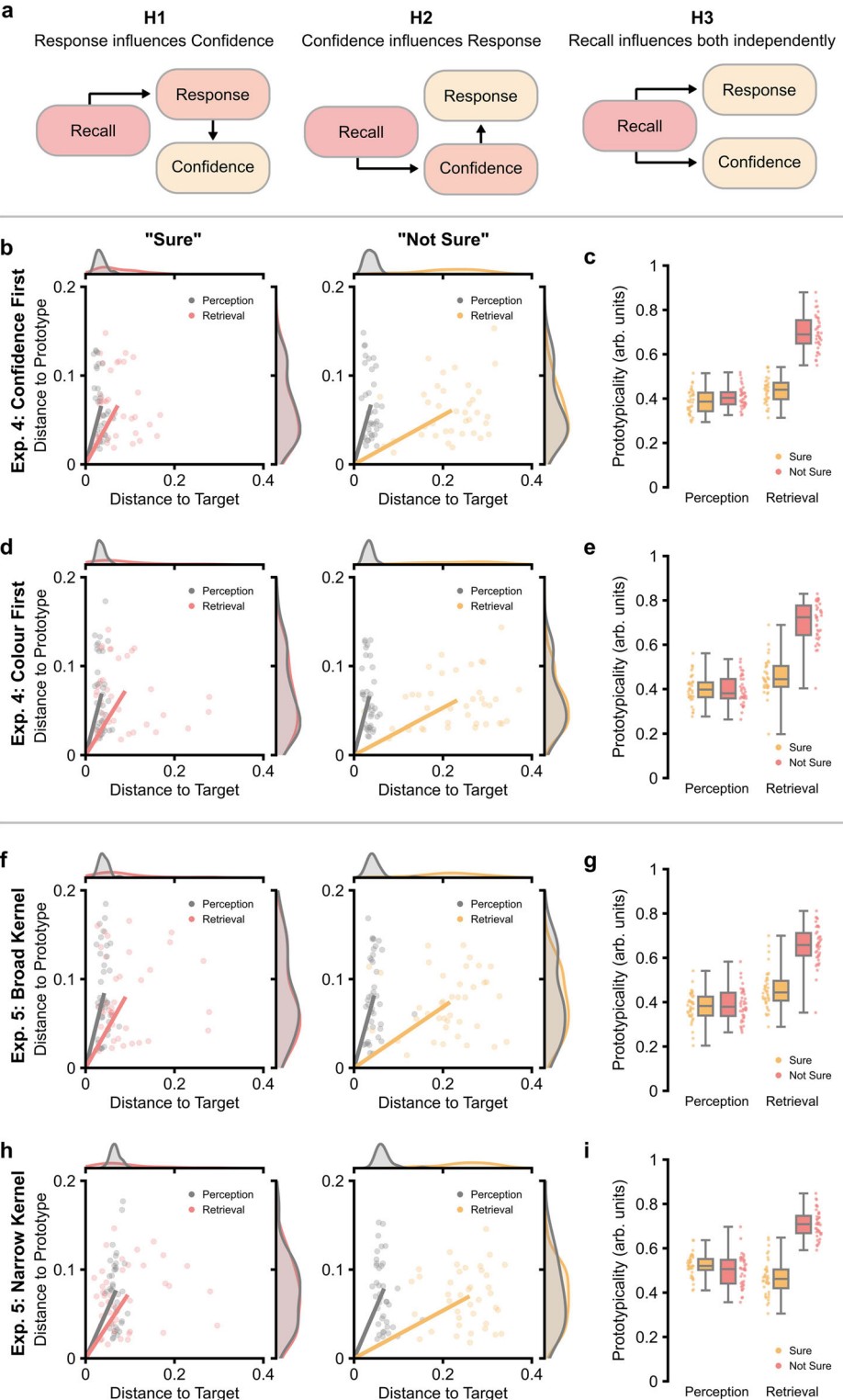

In Experiment 4, participants were often confident in their responses (mean: 58.4%; SEM: 3.3%) and a three-factor repeated measures ANOVA replicated the main effects for epoch [$F(1, 35) = 160.00$, $p < 0.001$, $\eta_P^2 = 0.82$, 95% CI = (0.69, 0.88)], confidence [$F(1, 35) = 160.84$, $p < 0.001$, $\eta_P^2 = 0.82$, 95% CI = (0.69, 0.88)], and an interaction between the two [$F(1, 35) = 233.17$, $p < 0.001$, $\eta_P^2 = 0.87$, 95% CI = (0.77, 0.91)] (see Fig. 3b–e). However, the order in which participants made their decisions did not produce a main effect on the prototypicality of responses [$F(1, 35) = 1.30$, $p = 0.261$, $\eta_P^2 = 0.04$, 95%

CI = (0.00, 0.21)], nor did we observe any interaction between retrieval order and the other factors [order*epoch: $F(1, 35) = 0.41$, $p = 0.528$, $\eta_P^2 = 0.01$, 95% CI = (0.00, 0.15); order*confidence: $F(1, 35) = 1.49$, $p = 0.231$, $\eta_P^2 = 0.04$, 95% CI = (0.00, 0.22); order*epoch*confidence: $F(1, 35) = 0.40$, $p = 0.534$, $\eta_P^2 = 0.01$, 95% CI = (0.00, 0.15)]. A Bayesian ANOVA indicated that confidence did not significantly influence prototypical responding (effect of task order: $BF_{10} = 0.23$; order*epoch: $BF_{10} = 0.27$; order*confidence: $BF_{10} = 0.29$; order*epoch*confidence: $BF_{10} = 0.13$).

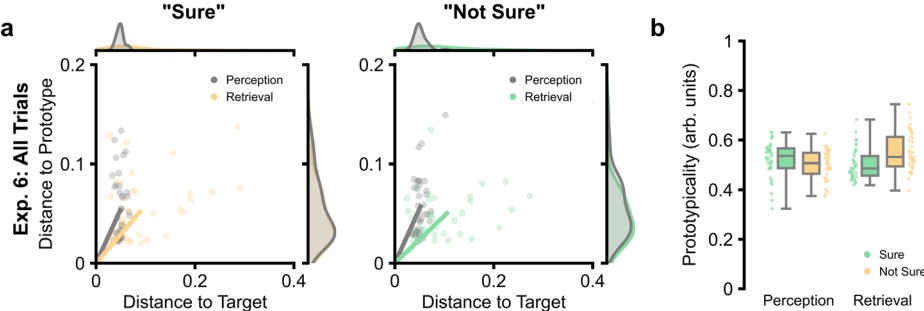

**Fig. 4 | Task performance in Experiment 6. a** Plots of task performance visualised in 2-dimensional space (*n* = 33 participants). Line plots represent the vector for each condition (grey: perception; coloured: retrieval). The density plots above and to the right of the main plots represent the distributions for each dimension separately. **b** Boxplots depicting prototypicality given epoch (x-axis) and confidence rating (hue). The boxplots display the median and interquartile range, with the whiskers capturing the range of the data. Individual dots reflect individual participants (*n* = 33 participants). All data reflects within-participant variation (Loftus & Masson, 1994).

With Experiment 4 providing little support for the first two hypotheses, we moved on to the last hypothesis, asking whether confidence is derived from the prototypicality of the memory trace itself. To test this idea, Experiment 5 manipulated the underlying distribution of the colour samples such that half the samples came from "broad" distributions while the other half came from "narrow" distributions. We reasoned that accurate responses for object-colour associations drawn from the narrow distributions would nonetheless appear prototypical as the absolute distance to the prototype would be small, and this would undermine confidence.

In line with the experiments above, participants were confident in their responses (mean: 61.5%; SEM: 2.5%). Furthermore, we saw the typical pattern of results for epoch, confidence and their interaction [epoch main effect: $F(1, 39) = 74.58$, $p < 0.001$, $\eta_p^2 = 0.66$, 95% CI = (0.46, 0.76); confidence main effect: $F(1, 39) = 163.08$, $p < 0.001$, $\eta_p^2 = 0.81$, 95% CI = (0.68, 0.86); epoch*confidence interaction: $F(1, 39) = 186.77$, $p < 0.001$, $\eta_p^2 = 0.83$, 95% CI = (0.71, 0.88); see Fig. 3f–i]. Critically, we found that distribution exerted a strong influence over prototypicality with (i) a main effect suggesting that narrow kernels lead to more prototypical responses across all conditions [$F(1, 39) = 63.48$, $p < 0.001$, $\eta_p^2 = 0.62$, 95% CI = (0.41, 0.73)]; (ii) an interaction with the epoch factor indicating that narrow kernels reduced the difference in prototypical responding between perception and retrieval [$F(1, 39) = 24.44$, $p < 0.001$, $\eta_p^2 = 0.39$, 95% CI = (0.15, 0.56)], and (iii) a three-way interaction between all factors [$F(1, 39) = 10.10$, $p = 0.003$, $\eta_p^2 = 0.21$, 95% CI = (0.03, 0.40)], suggesting that narrow kernels produced as smaller difference in confidence effects between perception and retrieval. No interaction was observed between confidence and kernel size [$F(1, 39) = 0.08$, $p = 0.783$, $\eta_p^2 < 0.01$, 95% CI = (0.00, 0.10)]. We then tested whether the proportion of "Sure" responses changed as a function of distribution. A paired samples t-test found that confidence was lower when recalling associations belonging to narrow kernels relative to broad kernels (mean number of "Sure" responses for broad kernels: 63.8%, standard deviation: 16.9%; mean response for narrow kernels: 59.2%, standard deviation: 16.3%; $t_{39} = 3.82$, $p < 0.001$, Cohen's $d = 0.61$, 95% CI = [0.27, 0.95]). The suggests that confidence may be derived from the prototypicality of the memory trace itself.

**Prototype-based distortion begins at encoding**
Lastly, given that confidence appears to be derived from prototype-based distortions within the memory trace itself, we asked whether the link between confidence and prototype-based distortion may extend to perception/encoding.

To test this idea, we first asked whether there is a relationship between prototype-based distortions observed during perception and those observed during recall. Indeed, we found a positive correlation across participants suggesting that the prototypicality of a perceptual judgement predicts the prototypicality of response for the same stimulus during recall (Exp. 1: $t_{44} = 5.72$, $p < 0.001$, Cohen's $d = 0.83$, 95% CI = [0.49, 1.17]; Exp. 2: $t_{33} = 4.49$, $p < 0.001$, Cohen's $d = 0.77$, 95% CI = [0.38, 1.15]; Exp. 3:

$t_{27} = 8.10$, $p < 0.001$, Cohen's $d = 1.58$, 95% CI = [1.00, 2.13]; Exp. 4: $t_{36} = 3.63$, $p = 0.001$, Cohen's $d = 0.60$, 95% CI = [0.19, 0.99]; Exp. 5: $t_{39} = 3.03$, $p = 0.004$, Cohen's $d = 0.48$, 95% CI = [0.15, 0.81]). Importantly, as our approach to computing memory-based prototypicality excludes preexisting perceptual distortions (see methods), this suggests that prototype-based distortions at perception not only persist through to recall but become exacerbated.

We then investigated whether confidence during perception maps onto distortions during recall. In Experiment 6, we asked participants to make a prospective confidence judgement immediately after perceiving/encoding the stimulus, explicitly asking them how confident they felt in their ability to recall the later pairing ("Sure" responses were equivalent to the previous experiments: mean: 62.5%; SEM: 3.6%). Here, a two-factor model continued to reveal a main effect for confidence, where prototypicality was lower for "Sure" responses [prototypicality for "Sure" responses: 0.51; prototypicality for "Not Sure" responses: 0.53; confidence main effect: $F(1, 36) = 4.46$, $p = 0.042$, $\eta_p^2 = 0.11$, 95% CI = (0.00, 0.31)]. While no main effect was observed for epoch [$F(1, 36) = 0.66$, $p = 0.442$, $\eta_p^2 = 0.02$, 95% CI = (0.00, 0.17)], we continued to observe the interaction between confidence and epoch, where the difference in prototypicality between confidence ratings was larger during retrieval than during perception [interaction term $F(1, 36) = 22.11$, $p < 0.001$, $\eta_p^2 = 0.38$, 95% CI = (0.14, 0.56); $\Delta$ prototypicality at perception ("Sure" > "Not Sure"): 0.02; $\Delta$ prototypicality at retrieval ("Sure" > "Not Sure"): −0.05; see Fig. 4a, b]. Altogether, these results suggest that we possess prospective awareness of the prototype-based distortions that we will encounter during later recall.

Given that both perceptual distortion and prospective confidence correlate with retrieval-related distortion, it is a possibility that prospective confidence does not directly predict memory-based distortion but instead relates to perceptual distortion, which in turn predicts memory-based distortion. To rule out this possibility, we conducted a multiple regression to quantify how well perceptual distortion and confidence can explain memory-based distortion as statistically independent factors. In this model, we found that prospective confidence continued to predict memory-based distortion ($t_{36} = -4.35$, $p < 0.001$, Cohen's $d = -0.72$, 95% CI = [−1.08, −0.36]; though perceptual distortion did not: $t_{36} = 1.56$, $p = 0.127$, Cohen's $d = 0.26$, 95% CI = [−0.07, 0.59]), suggesting the link between prospective confidence and memory-based distortion is not simply an indirect effect relating to perceptual distortion.

**Prototypical guessing and low confidence**
One remaining explanation for the link between confidence and prototypicality is that of prototypical guessing: when participants entirely forget an association, they may resort to making highly prototypical, guessed responses. While we attempted to capture this using subjective reporting, many participants reported few if any guesses (see Methods). Therefore, as an alternative approach to rule out prototypical guessing, we restricted our analyses to responses that fell between the target location and the nearest

prototype to that target location. This would help exclude prototypical guesses, which should fall on any prototype with equal probability. Using this approach, all analyses produced synonymous results to those presented above (see Supplementary Table 1), suggesting that prototypical guessing does not drive the link between confidence and prototypicality.

## Discussion

Prototypes exert a strong influence over memory; while they can help build a coherent narrative around key details stored in an episodic memory trace, they can also distort the memory when the episodic details are limited[23,25]. It remains unclear, however, whether we are metacognitively aware of when a memory is influenced by a prototype. Across six experiments, we find that prototypical responses at retrieval are accompanied by a reduction in self-reported confidence. Our results point towards an interpretation in which the prototypicality of a memory trace undermines confidence, rather than the lack of confidence driving prototypical reconstruction (as in some Bayesian models; e.g.[5,32]). Moreover, we demonstrate that this metacognitive awareness can emerge when we form a memory. Altogether, these results suggest that we hold a metacognitive awareness of the veracity of our memories.

Our results demonstrate that confidence negatively correlates with memory prototypicality and that the degree of prototypicality seems to lead to a decline in confidence. A recent study uncovered a similar correlation between confidence and prototypical distortions[31], though was unable to distinguish whether this confidence signal was derived from the prototypicality of the memory trace itself, or instead reflected a generically weak memory trace, which drove the use of prototypes during reconstruction (e.g.[5,32]). Our results favour the former. In Experiment 5, we manipulated stimuli to enhance/reduce their prototypicality and observed that more prototypical stimuli were recalled with less confidence. This suggests that confidence is derived from the prototypicality of a memory trace itself.

We also found that we first become metacognitively aware of prototype-based distortions during memory formation (see Experiment 6), with prospective confidence judgements made at perception correlating with prototype-based distortions at retrieval. This was statistically independent of the correlation between prototypicality links between perception and retrieval, ruling out the possibility of a proxy effect where confidence ratings correlated with perceptual prototypicality, which in turn correlated with recall prototypicality. This extends the findings of numerous studies demonstrating that prototype-based distortions first arise during encoding[35,38–44] to incorporate elements of metacognitive awareness. The fact that metacognitive awareness of prototype-based distortions can be observed as early as encoding further questions whether the confidence signal is purely linked to reconstruction-related phenomenon, and instead favours the idea that we are metacognitively aware of distortions of the memory trace itself.

To date, most studies have explored the effects of prototypes on memory using recognition tests (e.g.[45–47]; however, see also[48,49]), using methods such as old/new judgements (e.g.[23]) or multi-stimulus forced choice (e.g.[8,17]). These approaches perhaps dominate the field as they allow explicit control of the prototypicality of the test stimuli. However, there are many examples of when one cannot rely on recognition to accomplish a memory-dependent task (e.g., recounting our day, navigating the commute to work), questioning whether prototypicality has a similar influence over recall as it does on recognition. Indeed, our results suggest that such an assumption is erroneous (see also[31]). Numerous recognition studies have found that participants will often falsely recognise a prototype with high confidence when they never studied the prototype[21,50,51]. In contrast, we demonstrate a reduction in confidence when recalling prototypical memories, suggesting an inverted effect of prototypes on confidence during recall relative to recognition. While we cannot offer a definitive answer for this inversion, answers may be found in the different ways in which prototypes contribute to recall and recognition. When presented with a prototypical lure in a recognition test, its prototypical nature may elicit feelings of familiarity which aid recognition and lead to confident responses[52]. In contrast, when a cue is presented in a recall test and the recalled stimulus appears prototypical, the inability to distinguish it from stored prototypes undermines confidence[53]. While this interpretation remains speculative, the integration of our results with existing work on recognition memory suggests that prototypes have distinct effects on recall and recognition tests of memory. This may prove to be a critical distinction for those using confidence ratings as a proxy for memory veracity in several applied settings.

## Applications

Our demonstration of metacognitive awareness surrounding the veracity of recalled memories has important ramifications for eyewitness testimony[54,55]. Undoubtedly, eyewitness testimonies hinge upon the reliability of the witness[56], and while accuracy often positively correlates with confidence in these instances[56–59], the confidence-accuracy relationship can break down when presented with an external source of misleading information[26,60,61]. While this lack of metacognitive awareness regarding external sources of bias has been mitigated through the use of specialised interviews (e.g.[62,63]), it is unclear whether such interviews mitigate the effects of internal sources of bias (e.g., from prototypes) given the difficulty in quantifying such biases. Fortunately, our results demonstrate that, when recalling an event, individuals possess metacognitive awareness of prototypic distortions in memory, suggesting that the veracity of an eyewitness report is unlikely to be unintentionally biased by internal prototypes.

While our results demonstrate that young adults possess robust metacognitive awareness of prototype-driven biases in memory, it remains unclear whether similar effects persist in older adults. Indeed, numerous studies have shown that confidence-accuracy correlations decline in older age[64,65], with older adults more likely to confidently (but incorrectly) recognise lures[66,67], suggesting that metacognitive awareness of memory veracity may decline with age. However, these individuals also exhibit a general decline in confidence during memory tests[64,68] and, outside of the lab, subjective memory complaints can predict the later diagnosis of a memory-related disorder[69,70], suggesting that the relationship between confidence and memory accuracy in older adults is multifaceted. Consequently, it will be of interest to see how the correlation between confidence and prototype-based distortion predicts age-related mnemonic decline.

## Limitations

Here, we did not use a spectrally-constrained colour bar where we controlled the subjective psychological distance between each colour (e.g., Maximum Likelihood Difference Scaling[71]), but rather elected to use a spectrally-broad and (arguably) more ecologically valid colour bar. This decision was driven by the desire to tap into pre-existing, participant-specific colour prototypes rather than rely on participants building task-specific prototypes through lengthy training. Furthermore, this approach also freed us from the assumption that all participants perceive colour equally[72], which cannot be said for experimenter-defined prototypes derived from controlled, constrained colour bars. That said, the controlled, spectrally-constrained colour bar does have one key advantage over the colour bar we used. Specifically, it allows one to assume a linear relationship between response error and representation error. That is, a perceptual error that leads to a response that is 100 pixels away from the target can be assumed to be twice the error perceptual error that leads to a response that is 50 pixels away. Our task lacks this control and there may be representational "leaps" between continuous hues of the colour bar. We concede that such leaps may introduce noise into our measurements that suppresses the magnitude of the report effects. Indeed, when participants learned locations rather than colours (see Experiment 3), the magnitude of the observed effects rose drastically. Altogether, this suggests that our choice of colour bar did not bias our results towards a false positive and, if anything, may have suppressed more marginal effects.

Our analyses assume that participants have no prior associations between the objects and the colours in which they are presented. However, several studies have demonstrated that pre-existing associations do bias how

colour for an object is perceived and recalled[16,73,74]. Speculatively, this may explain why responses for object-colour associations (Experiments 1–2, 4–6) are more prototypical than responses for object-location associations (Experiment 3), as this prior knowledge biases colour choices more so than for the equivalent location choices. However, our observation of metacognitive effects for both colour- and location-cued recall suggests that canonical object-colour associations are not a major confounding factor. Nonetheless, future studies may benefit from accounting for canonical associations and including them as covariates to fully decorrelate their influence on behavioural responses.

Similarly, canonical colour knowledge may also provide an alternative explanation for differences in responses in Experiment 5 (where participants saw colours from narrow and broad kernels). Stimulus colours taken from the broad kernels are more likely to transcend canonical colour boundaries, helping participants to differentiate stimuli from the same kernel and boosting overall confidence in the response[75]. This could explain why we observed more "Sure" responses for colours taken from the broad kernel. However, colour boundaries would only introduce non-directional noise in prototypicality analyses, meaning it cannot explain why response prototypicality is consistently different between the two kernel widths. Again, this demonstrates how future research may benefit from detailed documentation of prior colour knowledge, particularly when using researcher-defined prototypes.

Here, we have focused on how exemplars are drawn to prototypes as this made up the majority of responses (76.8%; with the remainder being attributed to either repulsion from prototypes or total forgetting). However, other studies have found repulsion to play a larger role in prototype-based distortion (e.g.[18]). We speculate that this is due to differences in experimental design, particularly regarding stimulus competition. For example, when colour is the only dimension in which two exemplars differ, repulsion helps distinguish the stimuli (see[18]; similar competition-driven repulsion can also be observed on a neural level[76]). As our experiment did not involve competition between stimuli, we speculate that repulsion has no adaptive benefit here, perhaps explaining why little repulsion was observed.

## Conclusions

When we remember a past event, we reconstruct it by integrating recalled features with a variety of semantic, schematic, and prototypical details. To what extent, however, can we trust the veracity of memories that are recalled through such a process? We find that when our memories appear prototypical, we downweigh our confidence in them—an effect that we suggest is attributable to metacognitive awareness of prototype-based distortions of the memory traces themselves. These results contribute to addressing an existential question about the phenomenological experience of remembering and contribute to a number of applications, including eye-witness testimony and age-related cognitive decline.

## Data availability

The numerical dataset necessary to interpret, replicate and build upon the findings reported here is available in .csv format at https://osf.io/pmcqh/.

## Code availability

The analysis code (written in Python 3) is available at https://osf.io/pmcqh/.

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

## Acknowledgements

B.J.G. is funded by the Leverhulme Trust (https://www.leverhulme.ac.uk/ ; ECF-2021-628). The funders had no role in study design, data collection and analysis, decision to publish or preparation of the paper.

## Author contributions

C. K.: conceptualisation, methodology, formal analysis, writing—original draft;
Y. Z.: software, investigation, formal analysis, writing—review & editing;
B.J.G.: conceptualization, methodology, investigation, formal analysis, writing—original draft, supervision, project administration, funding acquisition.

## Competing interests

The authors declare no competing interests.
