## [Peer Review File · Communications Psychology]

7th Mar 24

Dear Dr Griffiths,

Thank you for your patience during the peer-review process. Your manuscript titled "Metacognitive awareness of memory distortion during recall" has now been seen by 3 reviewers, and I include their comments at the end of this message. They find your work of interest but raised some important points. We are interested in the possibility of publishing your study in Communications Psychology, but would like to consider your responses to these concerns and assess a revised manuscript before we make a final decision on publication.

We therefore invite you to revise and resubmit your manuscript, along with a point-by-point response to the reviewers. Please highlight all changes in the manuscript text file.

Editorially, we note that multiple reviewers expressed concerns about the strength of your evidence for the directional claim that the prototypicality of recall drives low confidence responses. Although you may attempt to more clearly articulate why you believe your data justify these causal claims, you should also tone down these claims and acknowledging the inferential limitations. Please also be sure to justify or reassess the appropriateness of pooling all responses (including the recall phase data) to define the prototypes. And your revisions should also clarify your study's unique contribution to the literature. Furthermore, the reviewers request more details needed about some of the methods so as to help readers better understand the paradigm, analysis procedures, and results presentation.

Please also include a statement regarding ethics and informed consent in the main manuscript, as per journal requirements.

I am attaching an Editorial Requests Table that details critical reporting requirements for the revised manuscript. Please attend to each item and ensure your manuscript is fully compliant. We are requesting that your manuscript aligns with these requirements as this facilitates the evaluation of your manuscript, reducing delays in re-review and potential future acceptance. If your revised manuscript is not aligned with these requests on major issues, such as those concerning statistics, it may be returned to you for further revisions without re-review. Additional information can be found in our style and formatting guide Communications Psychology formatting guide.

Please use the following link to submit your

- revised manuscript,
- point-by-point response to the referees' comments,
- cover letter (as a separate document),
- the Editorial Policy Checklist (see below),
- the Reporting Summary (see below), and
- the completed Editorial Request Table (attached):

[link redacted]

Best regards,

Jesse Rissman

Jesse Rissman, PhD

Editorial Board Member

Communications Psychology

orcid.org/0000-0001-8889-5539

REVIEWER EXPERTISE:

Reviewer #1: memory specificity and generalization

Reviewer #2: conceptual influences on episodic memory

Reviewer #3: metacognition and perceptual inference

REVIEWER REPORTS:

Reviewer #1 (Remarks to the Author):

SUMMARY

This manuscript presents a series of six experiments testing the relationship between prototype-based memory distortions and memory confidence. In five of the six experiments, participants learned object-color associations, indicating their perception of the to-be-learned color on a color scale at the time of learning. Memory for the color was later tested by presenting a black object and asking participants to pick its color from the color scale. Participants also indicated their confidence in their memory for the color. The sixth experiment was similar, but participants learned object-location associations rather than object-color associations. The authors used a k-means clustering approach to determine subject-level, subjective prototypes. They then tested whether color scale judgments tended to be more biased toward prototypes during retrieval compared to perception/encoding, and more biased for low confidence ('not sure' and 'guess') compared to high confidence ('sure') responses. They generally found that low confidence responses were more prototypical than high confidence responses, and more so in recall than in perception.

EVALUATION

Overall, I liked this paper. I thought the task was well designed, and the finding that prototypical responses were associated with lower confidence was an interesting departure from recognition findings. While I liked that the authors estimated subject-specific, subjective prototypes, I was not sure that pooling across all the data (perception, recall) to define those prototypes was appropriate. I was expecting prototypes to be defined based on the perception data and then applied to recall to test distortion from their perception at encoding. Another aspect of the study was the author's attempt to disentangle the direction of causality between confidence and prototypicality. After finding in Experiments 1-2 that lower confidence responses were also more prototypical, the authors sought to determine whether recalling a more prototypical color led to low confidence responses or if the low confidence drives participants to make a prototypical selection. I think this is an interesting question, but I did not find the manipulations to be very convincing in supporting the conclusion that it is the prototypicality of the recall that drives low confidence responses. I do not think they have ruled out the possibility that participants tend to make prototypical guesses when they have forgotten the specific color. Nonetheless, I think these experiments make an interesting contribution that would be of interest to a broad set of memory researchers.

SPECIFIC COMMENTS

1. In the third paragraph of the introduction, the authors describe known prototype effects in recognition memory tests (i.e., enhanced endorsement of never-seen prototypes). They then state, "However, it remains unclear whether we have a similar lack of metacognitive awareness of prototype-based interference when actively recalling our memories." This statement is making the simple point that it is not clear whether these prototype effects extend to recall tests, but it was not very clear to me on first reading that that was the takeaway point. Also, the next sentence about this raising 'an existential question about our mental autobiographies' seems like an overstatement of the problem.

2. Table 2 is a bit unwieldy and does not really work as a pseudo-visualization. Would it be possible to summarize the 1, 2, 3's listed for each experiment as columns in the table? For example, one of the columns could be 'distribution' and could list 'uniform' for most of the experiments but list 'two broad clusters, two narrow clusters' for experiment 4. There could be a column for domain, which would be 'color' for experiments 1-5 and 'location' for experiment 6. It seemed to me that much of the information summarized in this table was included through the manuscript, making it unnecessary to use the table to provide detailed information. But the table being so wordy means that it is not an effective 'cheat sheet' for the differences between experiments.

3. A small note: Table 2 refers to a training block, but it appears before lines 71-73 that explain what the training block is. It would be helpful to make sure the table appears after the necessary text to explain it.

4. I had several questions about the k-means clustering analysis. On the technical side, it would be helpful to give a 1-2 sentence conceptual description of what ‘optimal inertia’ and ‘k’s silhouette’ are since they are key metrics. I also wondered about the approach of pooling all responses. When I read line 78, it did not even register to me that the authors meant that they were estimating the number of clusters from both the perception and recall data simultaneously because I had just assumed that they would be defined from perception and applied to recall. I understand that this approach allows the authors to measure the effect of perception vs. retrieval, but it seems circular to let the memory distortion data influence the definition of the prototypes. Couldn’t including the recall data points lead to an underestimation of the memory distortion since the prototype could be pulled ‘closer’ to the distorted data in the k-means clustering analysis?

5. I think it would be helpful to make it explicit that, for most of the experiments, there was just one confidence rating that is being used to sort both the perception and retrieval trials. I was confused initially when both epoch and confidence rating were used as factors in the analysis as it made me question whether there was a confidence rating at perception (I realize that there is one for one of the experiments, but that comes up later).

6. Figure 2: A key x-axis label appears only at the bottom of the figure (“Distance to target”), in the E panel but not in the A panel. I understand not wanting to clutter the figure with redundant text, but it is confusing if you first look at the A panel and don’t see an axis label. It also seems like there is a best fit line for perception in the scatter plots, but the grey color of the line is not visible over the grey of the dots.

7. I did not find the mere manipulation of the order of the color judgment vs. confidence judgment (Experiment 3) very compelling as pointing toward prototypicality driving confidence, not vice versa. While I find the manipulation in Experiment 4 more interesting, I think it is hard to understand how participants experience differences in the underlying color distributions. The authors show that it affects the relationship between confidence and prototypicality, but I’m not sure the authors’ interpretation of the lack of a confidence effect for narrow distributions is necessarily right. For example, I wondered how broad distributions being more likely to cross color label category boundaries would affect participants’ perceptions. If anything, I think this experiment could be moved to the end as the one that is potentially most evocative but also most open to other interpretations.

8. Lines 214-215: I think it would be helpful to report the rates of confident responses separately for the narrow and broad kernels.

9. I also wondered whether there is something in the reaction time data that could help with the question about whether prototypical recall drives confidence or vice versa. My intuition would be that participants sometimes forget the precise color but still remember the color label. When that happens, they may simply make a guess based on the prototype. If this were the case, I would expect more prototypical judgments to be slower. How does reaction time relate to prototypicality? Does it differ for confident versus not confident responses?

10. Discussion: I found the ideas in the discussion to be a bit out of order. There are some very speculative paragraphs about neural substrates and the importance for real world applications before discussing how these findings differ from recognition findings on prototypicality. It then ends (prior to the summary) with a technical note about the color bar.

11. I was also disappointed that the authors did not really attempt to offer a reason why recall might be different from recognition with regards to influences of prototypicality. Instead, they make entirely more speculative comments about 'the ever-expanding hunt for the neural underpinnings of consciousness'. Essentially, I think the recall/recognition difference needs to be moved up in the discussion and needs to offer a more grounded reconciliation of findings.

Reviewer #2 (Remarks to the Author):

The authors conducted a series of behavioral experiments that investigated whether there exists a metacognitive awareness of the retrieval of memories that are more biased by a category prototype. In these experiments, the authors use self-reported confidence as a proxy for metacognitive awareness, and they assess confidence for the retrieval of color-object associations (and location-object associations in one study). Across all experiments, they find that less confident responses exhibit more bias in color memory in the direction of the participant's prototypical response. They confirm that reversing the direction of the color bar and switching the order of confidence and recall do not impact the prototypicality effects, manipulate responses to be less confident and more prototypical by restricting the kernel width of the clustered colors, and replicate their findings using object-location associations.

I found the topic of the paper to be well-written, well-motivated, and very timely, as it explores an under-studied element of the reconstructive memory theory. It inspired interesting thoughts about the exact source of meta-cognitive signals surrounding memory, and how these may differ from signals that we think correspond to recognition strength. However, some work is needed to specify the unique contribution to the literature that this paper makes given that a relationship between

confidence and biases in memory towards prototypes has already been demonstrated. There are also some specific sections that could benefit from improvement – namely, describing the angle measure of prototypicality in more depth and with more examples, and scaling back some of their interpretations regarding the directionality of the relationship between confidence and memory responses. Please see below for detailed comments.

Major comments

More detail about the prototypicality measure would be useful. How is the angle measure computed when a participant overshoots the prototype (e.g. recalled color for the apple is to the left of the prototype) or when a recalled color is repulsed from its prototype (e.g. recalled apple color is to the right of its perceived color)? It may be helpful to modify some of the examples pictured in Figure 1C to highlight these other responses.

In Experiment 3, two groups are tested to manipulate the order of the cued recall and self-reported confidence. There are no effects of task order or interactions with confidence and prototypicality. The authors interpret this as a more prototypical memory undermining confidence rather than the reverse, but a lack of an order effect could also mean that participants make both decisions (color and confidence) mentally before recording any response. It could also mean that neither task interfered with each other; in other words, making confidence responses first did not modulate prototypicality in memory, but the act of responding to the recall did not modulate confidence either. The conclusion about the directionality of these effects should be scaled back. I do think the manipulation of the kernel width in Experiment 4 is more effective at demonstrating the directionality between prototypical responses and confidence so this interpretation doesn't need to be completely removed.

Tompary and Thompson-Schill find a very similar pattern of results as is reported here: as self-reported confidence decreases, there is a corresponding increase in bias towards a category prototype for object-location associations (Tompary and Thompson-Schill 2023, Experiment 2). Please include this reference when discussing relevant background.

Biases in color memory can also arrive from prior knowledge of the canonical features of an item, like size or color - the color work is being done by Kimele Persaud. Given that some objects have canonical colors (red apple) and some don't, is there a chance that memory for some items may have influenced an over-reliance on its canonical color?

Work by the Kuhl lab on the repulsion of memories with highly similar features (including color hue) appears highly relevant and the paper would benefit from a discussion of how their approaches are related and why they give rise to diverging results.

Minor comments

It's difficult to use the figure and figure text alone to understand whether Distance to Target and Distance to Prototype are both using the Recalled Colour to compute the distances – I'd recommending adding this in the figure text to make it very clear.

The box plots for Retrieval in Figure 3A are shaded darker than all other box plots in that figure e – is this intentional? If so, please explain in the figure text.

Reviewer #3 (Remarks to the Author):

This study investigates the interesting question to what extent we are aware of distortions in episodic memory recall caused by semantic knowledge. Specifically, it remains unclear whether we are metacognitively aware of prototype-based inference when recalling memories. To this end, they developed a novel experimental paradigm in which participants memorize and later recall colour-object pairs. Prototypical colours are identified using K-means clustering, showing that participants to indeed collapse the colour space into “prototypes”. Results over 6 experiments consistently showed that more prototypical responses were associated with decreased confidence, indicating metacognitive awareness of prototypical biases. Several confounds are controlled for such as the location of the response, the type of stimulus (colour and location) and the order of responses and confidence ratings. This is an original study, that is technically sound and reveals interesting and novel findings that will be of interest to memory and metacognition researchers. I only have a few comments.

If I understood this correctly, the authors did test whether confidence during encoding predicted later memory bias but they did not investigate whether errors during encoding predicted later bias. If participants already show a prototypical effect for some objects during encoding, are those then also the objects that drive the later memory bias?

The results from experiment two reveal that samples from a narrower distributions lead to more prototypical responses, during both perception and retrieval. Could this not just be explained by

the fact these samples ARE more prototypical, based on the fact that the distances to the prototype are smaller by design? This would also explain why confidence does not seem to have much of an influence in this condition.

Minor comments

- I found the measure of prototypicality quite hard to understand. It might help to give a few examples of stimuli with high and low prototypicality scores here.

- Line 100: I wouldn't call a 60/40 split approximately even

EDITORIAL POLICIES

We ask that you ensure your manuscript complies with our editorial policies and reporting requirements.

To that end, we require revised manuscripts to be accompanied by two completed items: a reporting summary that collects information on study design and procedure, and an editorial policy checklist that verifies compliance with all required editorial policies.

Nature Research Reporting Summary

Editorial Policy Checklist

All points on the policy checklist must be addressed. Your revised manuscript can only be sent back to the referees if these checklists are completed and uploaded with the revision.

Notes: If you have submitted a Stage 1 Registered Report, Review, Primer, Comment, or Perspective you do not need to submit these forms. If you have already submitted these forms, you may disregard this request.

* TRANSPARENT PEER REVIEW: Communications Psychology uses a transparent peer review system. This means that we publish the editorial decision letters including Reviewers' comments to the authors and the author rebuttal letters online as a supplementary peer review file. However, on author request, confidential information and data can be removed from the published reviewer reports and rebuttal letters prior to publication. If your manuscript has been previously reviewed at another journal, those Reviewers' comments would not form part of the published peer review file.

Reviewer #1 (Remarks to the Author):

This manuscript presents a series of six experiments testing the relationship between prototype-based memory distortions and memory confidence. In five of the six experiments, participants learned object-color associations, indicating their perception of the to-be-learned color on a color scale at the time of learning. Memory for the color was later tested by presenting a black object and asking participants to pick its color from the color scale. Participants also indicated their confidence in their memory for the color. The sixth experiment was similar, but participants learned object-location associations rather than object-color associations. The authors used a k-means clustering approach to determine subject-level, subjective prototypes. They then tested whether color scale judgments tended to be more biased toward prototypes during retrieval compared to perception/encoding, and more biased for low confidence ('not sure' and 'guess') compared to high confidence ('sure') responses. They generally found that low confidence responses were more prototypical than high confidence responses, and more so in recall than in perception.

Overall, I liked this paper. I thought the task was well designed, and the finding that prototypical responses were associated with lower confidence was an interesting departure from recognition findings. While I liked that the authors estimated subject-specific, subjective prototypes, I was not sure that pooling across all the data (perception, recall) to define those prototypes was appropriate. I was expecting prototypes to be defined based on the perception data and then applied to recall to test distortion from their perception at encoding. Another aspect of the study was the author's attempt to disentangle the direction of causality between confidence and prototypicality. After finding in Experiments 1-2 that lower confidence responses were also more prototypical, the authors sought to determine whether recalling a more prototypical color led to low confidence responses or if the low confidence drives participants to make a prototypical selection. I think this is an interesting question, but I did not find the manipulations to be very convincing in supporting the conclusion that it is the prototypicality of the recall that drives low confidence responses. I do not think they have ruled out the possibility that participants tend to make prototypical guesses when they have forgotten the specific color. Nonetheless, I think these experiments make an interesting contribution that would be of interest to a broad set of memory researchers.

We would like to thank the reviewer for the thoughtful and thorough review. As the last point about prototypical guessing did not appear in the listed points, we address it here. We entirely agree with the point and have included supplementary analyses to address it. Specifically, in this analysis, we only consider responses that fall between the target location and its nearest prototype, by and large removing responses that reflect prototypical guessing following total forgetting. These analyses produce synonymous results to those presented in the main text, suggesting that the link between low confidence and prototypicality cannot be attributed to prototypical guessing (see quote below; see page 13, lines 328-337; supplementary table 1).

"One remaining explanation for the link between confidence and prototypicality is that of prototypical guessing: when participants entirely forget an association, they may resort to making highly prototypical, guessed responses. To rule out this possibility, we restricted our analyses to responses that fell between the target location and the nearest prototype to that target location, which by and large would exclude prototypical guesses (which should fall on any prototype with equal probability). Using this approach, all analyses produced synonymous results to

those presented above (see supplementary table 1), suggesting that prototypical guessing does not drive the link between confidence and prototypicality.”

1. In the third paragraph of the introduction, the authors describe known prototype effects in recognition memory tests (i.e., enhanced endorsement of never-seen prototypes). They then state, “However, it remains unclear whether we have a similar lack of metacognitive awareness of prototype-based interference when actively recalling our memories.” This statement is making the simple point that it is not clear whether these prototype effects extend to recall tests, but it was not very clear to me on first reading that that was the takeaway point. Also, the next sentence about this raising ‘an existential question about our mental autobiographies’ seems like an overstatement of the problem.

Thank you for highlighting this. We have rephrased this paragraph following suggested literature additions from Reviewer 2 and have incorporated the concerns above into the rewrite. We no longer dwell on recognition (this instead is moved to the discussion), but instead get straight to the point regarding recall (see quote; see page 2, lines 16-24).

“To what extent are we metacognitively aware of prototype-based distortions in memory? While an explicit question about prototypes may in and of itself induce metacognitive awareness, it is possible to probe metacognitive awareness implicitly using confidence ratings. Indeed, a recent study has done so, showing that when we actively recall memories that are distorted by prototypes, confidence declines [30]. However, it remains unclear whether this lack of confidence (i) reflects metacognitive awareness of prototype-based distortions embedded in a memory trace, or (ii) drives prototype-based distortions during reconstruction (e.g., [5,31]). Delineating these hypotheses would offer key insights into how and when prototype-based distortions influence memory, and whether these distortions can be disentangled from a memory trace in applied settings (e.g., eye-witness testimony).”

2. Table 2 is a bit unwieldy and does not really work as a pseudo-visualization. Would it be possible to summarize the 1, 2, 3’s listed for each experiment as columns in the table? For example, one of the columns could be ‘distribution’ and could list ‘uniform’ for most of the experiments but list ‘two broad clusters, two narrow clusters’ for experiment 4. There could be a column for domain, which would be ‘color’ for experiments 1-5 and ‘location’ for experiment 6. It seemed to me that much of the information summarized in this table was included through the manuscript, making it unnecessary to use the table to provide detailed information. But the table being so wordy means that it is not an effective ‘cheat sheet’ for the differences between experiments.

We have updated the table in accordance with the reviewer’s suggestions (see page 5).

3. A small note: Table 2 refers to a training block, but it appears before lines 71-73 that explain what the training block is. It would be helpful to make sure the table appears after the necessary text to explain it.

We have moved the table so that it follows its reference in the text (for in-text reference, see page 5, lines 79-82; for table on page 5).

4. I had several questions about the k-means clustering analysis. On the technical side, it would be helpful to give a 1-2 sentence conceptual description of what ‘optimal inertia’ and ‘k’s silhouette’ are since they are key metrics. I also wondered about the approach of pooling all responses. When I read line 78, it did not even register to me that the authors meant that they were estimating the number of clusters from both the

perception and recall data simultaneously because I had just assumed that they would be defined from perception and applied to recall. I understand that this approach allows the authors to measure the effect of perception vs. retrieval, but it seems circular to let the memory distortion data influence the definition of the prototypes. Couldn't including the recall data points lead to an underestimation of the memory distortion since the prototype could be pulled 'closer' to the distorted data in the k-means clustering analysis?

We have included additional detail on the two methods and have also included a link to their implementation so interested readers can fully explore how the algorithms run (see quote; see pages 6, lines 100-104).

"Optimal inertia refers to the iteration which produces the smallest sum of all squared distances between data points within a cluster and the cluster centroid. The optimal number of clusters was then defined as the k with the largest mean silhouette score. The silhouette score describes how well one cluster is separated from others; a high silhouette score indicates that good separation between clusters. For full details, see the scikit-learn API."

We understand the reviewer's confusion about the cluster estimation approach and appreciate their suggested alteration. However, the nature of the experimental design precludes any method that defines prototypes based solely on the perceptual data. As most experiments had stimulus colours placed equidistantly along the colour bar, and participants performed exceptionally well during perception (mean response error: 0.04, where 1 represents the length of the entire colour bar), the k-means algorithms will inevitably perform poorly on this data alone. In other words, the prototypes only really become apparent in the recall data. That said, we do believe that the reviewer's point on circularity is correct and have found an alternative solution: *cross-validation*. For every response, the prototypicality of the response was computed using clusters defined by all data points *except* this individual response (known as the "leave-one-out" approach to cross-validation). Using this approach, the individual response can no longer influence the cluster definitions, circumventing concerns that we underestimate memory distortion. The change in analytical approach has led to all analyses being re-run. No major change has been encountered aside from an interaction in Experiment 6 now passing the threshold for statistical significance (for full details, see lines 270-302). We have updated the methods section to describe the change in approach (see quote below; see page 6, lines 95-99).

"Prototypes were derived from the data for each participant individually using cross-validated (here, leave-one-out) k-means clustering. For every response of every participant, we pooled all responses, excluding the response of interest, and derived k clusters, with k iteratively taking all integer values between 2 and 10 (inclusively). For each value of k, the k-means clustering algorithm was run 300 times, with the cluster centroids defined as the series of 10 runs of these runs which produced optimal inertia."

5. I think it would be helpful to make it explicit that, for most of the experiments, there was just one confidence rating that is being used to sort both the perception and retrieval trials. I was confused initially when both epoch and confidence rating were used as factors in the analysis as it made me question whether there was a confidence rating at perception (I realize that there is one for one of the experiments, but that comes up later).

We have made it explicit that participants provided a confidence rating once per stimulus, and that this occurs during retrieval (except for Experiment 6, which occurs during encoding; see quote below; see page 5, see lines 79-82).

“In all experiments, participants provided a confidence rating once per stimulus. In most experiments, this was during memory retrieval; the exception was Experiment 6, where a prospective confidence rating was collected during memory encoding (see Table 2).”

6. Figure 2: A key x-axis label appears only at the bottom of the figure (“Distance to target”), in the E panel but not in the A panel. I understand not wanting to clutter the figure with redundant text, but it is confusing if you first look at the A panel and don’t see an axis label. It also seems like there is a best fit line for perception in the scatter plots, but the grey color of the line is not visible over the grey of the dots.

Agreed. We have added the x-label to all figure panels in all figures. We have also recoloured the plots to better accentuate the line of the mean resultant vectors.

7. I did not find the mere manipulation of the order of the color judgment vs. confidence judgment (Experiment 3) very compelling as pointing toward prototypicality driving confidence, not vice versa. While I find the manipulation in Experiment 4 more interesting, I think it is hard to understand how participants experience differences in the underlying color distributions. The authors show that it affects the relationship between confidence and prototypicality, but I’m not sure the authors’ interpretation of the lack of a confidence effect for narrow distributions is necessarily right. For example, I wondered how broad distributions being more likely to cross color label category boundaries would affect participants’ perceptions. If anything, I think this experiment could be moved to the end as the one that is potentially most evocative but also most open to other interpretations.

This is an interesting point, and one related to that of Reviewer 2 who commented on existing canonical object-colour associations. Colour boundaries may help differentiate stimuli from the same kernel, and therefore boost confidence during recall (e.g. Watier & Collin, 2012, Memory). This may well explain why we saw more “Sure” responses for broad distributions (relative to narrow distributions). However, these boundaries would only introduce noise into the prototypicality analyses, and hence suppress any differences between the conditions. This is not what we observed, suggesting it cannot explain all facets of the results of the original Experiment 4 (now Experiment 5). Nonetheless, it is an important point, and we have included it in the discussion (see quote below; see page 16, see lines 436-444).

“Similarly, canonical colour knowledge may also provide an alternative explanation for differences in responses in Experiment 5 (where participants saw colours from narrow and broad kernels). Stimulus colours taken from the broad kernels are more likely to transcend canonical colour boundaries, helping participants to differentiate stimuli from the same kernel and boosting overall confidence in the response [76]. This could explain why we observed more “Sure” responses for colours taken from the broad kernel. However, colour boundaries would only introduce non-directional noise in prototypicality analyses, meaning it cannot explain why response prototypicality is consistently different between the two kernel widths. Again, this demonstrates how future research may benefit from detailed documentation of prior knowledge, particularly when using researcher-defined prototypes”.

We considered the suggestion to move this experiment to the end. However, we found that it made the narrative structure more confusing to follow. That said, we have moved the spatial experiment to the first results section so that, generally speaking, the more evocative experiments come towards the end of the section.

8. Lines 214-215: I think it would be helpful to report the rates of confident responses separately for the narrow and broad kernels.

Agreed, we have included the mean number of confident responses. They accompany the related inferential statistics (see quote below; see pages 11-12, lines 280-284).

“A paired samples t-test found that confidence was lower when recalling associations belonging to narrow kernels relative to broad kernels [mean number of “Sure” responses for broad kernels: 63.8% (standard deviation: 16.9%); mean response for narrow kernels: 59.2% (standard deviation: 16.3%); $t_{39} = 3.82, p < 0.001$ ”

9. I also wondered whether there is something in the reaction time data that could help with the question about whether prototypical recall drives confidence or vice versa. My intuition would be that participants sometimes forget the precise color but still remember the color label. When that happens, they may simply make a guess based on the prototype. If this were the case, I would expect more prototypical judgments to be slower. How does reaction time relate to prototypicality? Does it differ for confident versus not confident responses?

This is an intriguing hypothesis. To test this, as suggested, we correlated response time (when making the colour judgment) with the prototypicality of responses for each participant individually, then used a one-sample t-test to examine whether there was a consistent trend in correlation across participants. In Experiment 1, we found no effect for perceptual nor retrieval responses when correlating across all trials (perception: $t(44) = 0.35, p = 0.726$; retrieval: $t(44) = -1.44, p = 0.158$). When splitting retrieval trials by confidence, confident responses showed an effect just passing the threshold for significance (“Sure”: $t(44) = -2.02, p = 0.049$; “Unsure”: $t(44) = -0.14, p = 0.888$). We could not replicate this effect in Experiment 2 (“Sure”, $t(33) = -0.17, p = 0.867$; $t(33) = 0.66, p = 0.513$). Given the marginal significance ($p = 0.049$) in Experiment 1, the failure to replicate, and the fact that the observed correlation was the inverse of that hypothesis proposed by the reviewer, we feel that there is insufficient evidence to definitively conclude an effect (or lack thereof) between prototypicality and response times. We have mentioned this in the methods of the main text (see quote below; see page 7, lines 147-149) and included a supplementary figure (see Supplementary Figure 1; see below).

“Lastly, following a reviewer’s suggestion, we explored whether reaction time fluctuates as a function of prototypicality. However, we found no significant relationship between the two variables (for further details, see supplementary figure 1).”

10. Discussion: I found the ideas in the discussion to be a bit out of order. There are some very speculative paragraphs about neural substrates and the importance for real world applications before discussing how these findings differ from recognition findings on prototypicality. It then ends (prior to the summary) with a technical note about the color bar.

Thank you for highlighting this. We have now moved the discussion on recall vs. recognition to the first section of the discussion (see pages 14-15, lines 368-386). We have also removed the section on the neural substrates this was a speculative point to which our purely behavioural data could not contribute substantially. We have kept the *Methodological Considerations* section at the end of the discussion as we feel it is important to acknowledge design details that might influence the results of any experiment that aims to expand upon our work.

11. I was also disappointed that the authors did not really attempt to offer a reason why recall might be different from recognition with regards to influences of prototypicality. Instead, they make entirely more speculative comments about ‘the ever-expanding hunt for the neural underpinnings of consciousness’. Essentially, I think the recall/recognition difference needs to be moved up in the discussion and needs to offer a more grounded reconciliation of findings.

Many thanks for the suggestion. We have moved the paragraph on recall vs. recognition to the first section of the Discussion and added a more grounded and detailed summation of possible explanations (see quote below; see pages 14-15, see lines 378-386). It nonetheless remains somewhat speculative as we are not aware of any study exploring the intersection between prototypes, confidence, and recall vs. recognition, but we hope that our ideas perhaps provide more tangible ideas for future research.

“While we cannot offer a definitive answer for this inversion, answers may be found in the differential ways in which prototypes contribute to recall and recognition. When presented with a prototypical lure in a recognition test, its prototypical nature may elicit feelings of familiarity which aid recognition and lead to confident responses [54]. In contrast, when a cue is presented in a recall test and the recalled stimulus appears prototypical, the inability to distinguish it from stored prototypes undermines confidence [55]. While this interpretation remains speculative, the integration of our results with existing work on recognition memory suggests that prototypes have distinct effects on recall and recognition tests of memory.”

Reviewer #2 (Remarks to the Author):

The authors conducted a series of behavioral experiments that investigated whether there exists a metacognitive awareness of the retrieval of memories that are more biased by a category prototype. In these experiments, the authors use self-reported confidence as a proxy for metacognitive awareness, and they assess confidence for the retrieval of color-object associations (and location-object associations in one study). Across all experiments, they find that less confident responses exhibit more bias in color memory in the direction of the participant's prototypical response. They confirm that reversing the direction of the color bar and switching the order of confidence and recall do not impact the prototypicality effects, manipulate responses to be less confident and more prototypical by restricting the kernel width of the clustered colors, and replicate their findings using object-location associations.

I found the topic of the paper to be well-written, well-motivated, and very timely, as it explores an under-studied element of the reconstructive memory theory. It inspired interesting thoughts about the exact source of meta-cognitive signals surrounding memory, and how these may differ from signals that we think correspond to recognition strength. However, some work is needed to specify the unique contribution to the literature that this paper makes given that a relationship between confidence and biases in memory towards prototypes has already been demonstrated. There are also some specific sections that could benefit from improvement – namely, describing the angle measure of prototypical in more depth and with more examples, and scaling back some of their interpretations regarding the directionality of the relationship between confidence and memory responses. Please see below for detailed comments.

We would like to thank the reviewer for their thoughtful and incisive comments. Their highlighting of Tompary and Thompson-Schill's research has been a profound help in contextualising and clarifying the unique contribution of our manuscript to the literature (see point 3).

Major comments

1. More detail about the prototypicality measure would be useful. How is the angle measure computed when a participant overshoots the prototype (e.g. recalled color for the apple is to the left of the prototype) or when a recalled color is repulsed from its prototype (e.g. recalled apple color is to the right of its perceived color)? It may be helpful to modify some of the examples pictured in Figure 1C to highlight these other responses.

In short, the method classifies repulsed responses as being very low in prototypicality and overshoots as very high in prototypicality. Due to limited space, we have only been able to visualise the repulsion in figure panel, but have described both in the figure legend (see page 4 for figure 1; see quote below for added legend text).

"In instances when a response is repulsed by a prototype (see blue truck), prototypical bias is very low. In instances when a response overshoots its prototype (not visualised), prototypical bias is very high."

The original analyses included both repulsions and overshoots, though the reviewer's suggestion did make us question whether the inclusion of repulsion or overshoots influenced the main results. A control analysis that excluded all responses that were either repulsions or overshoots suggested they did not, as this analysis produced near identical results to the original analyses. We have included these additional analyses in

the supplementary materials (see supplementary table 1) and referenced them in the figure (see quote below; see legend on page 4).

“Excluding repulsions and overshoots has no impact on the central results (see supplementary table 1).”

2. In Experiment 3, two groups are tested to manipulate the order of the cued recall and self-reported confidence. There are no effects of task order or interactions with confidence and prototypicality. The authors interpret this as a more prototypical memory undermining confidence rather than the reverse, but a lack of an order effect could also mean that participants make both decisions (color and confidence) mentally before recording any response. It could also mean that neither task interfered with each other; in other words, making confidence responses first did not modulate prototypicality in memory, but the act of responding to the recall did not modulate confidence either. The conclusion about the directionality of these effects should be scaled back. I do think the manipulation of the kernel width in Experiment 4 is more effective at demonstrating the directionality between prototypical responses and confidence so this interpretation doesn’t need to be completely removed.

Yes, we share the reviewer’s view here, though did not articulate it sufficiently clearly in our initial submission. We have revised the title of that section in the Results to scale back the claim (see page 10, lines 220-221), added a clarifying paragraph at the beginning of this section (see quote below; see page 10, lines 222-236), added a hypothesis panel to Figure 3 to visualise the multiple possible explanations, and made minor changes to phrasing throughout the manuscript to dial back directional claims.

“Confident in the fact that prototype-based distortions correlate with a reduction in confidence, we then asked why this occurs. We considered three different hypotheses (see figure 3A): (i) confidence is derived from the similarity between a response and the prototype [37]; (ii) weak memory traces result in a lack of confidence which, in turn, leads participants to make prototypical reconstructions/responses [5]; (iii) confidence is unrelated to the response and is derived from the prototypicality of the memory trace itself”

3. Tomparý and Thompson-Schill find a very similar pattern of results as is reported here: as self-reported confidence decreases, there is a corresponding increase in bias towards a category prototype for object-location associations (Tomparý and Thompson-Schill 2023, Experiment 2). Please include this reference when discussing relevant background.

It was to our detriment that we overlooked this paper. We have rewritten the penultimate paragraph of the introduction to acknowledge the work and elaborate on how our experiments build upon it (see quote below; see page 2, lines 16-24).

“To what extent are we metacognitively aware of prototype-based distortions in memory? While an explicit question about prototypes may in and of itself induce metacognitive awareness, it is possible to probe metacognitive awareness implicitly using confidence ratings. Indeed, a recent study has done so, showing that when we actively recall memories that are distorted by prototypes, confidence declines [30]. However, it remains unclear whether this lack of confidence (i) reflects metacognitive awareness of prototype-based distortions embedded in a memory trace, or (ii) drives prototype-based distortions during reconstruction (e.g., [5,31]). Delineating these hypotheses would offer key insights into how and when prototype-based distortions influence memory, and whether these distortions can be disentangled from a memory trace in applied settings (e.g., eye-witness testimony).”

4. Biases in color memory can also arrive from prior knowledge of the canonical features of an item, like size or color - the color work is being done by Kimele Persaud. Given that some objects have canonical colors (red apple) and some don't, is there a chance that memory for some items may have influenced an over-reliance on its canonical color?

We agree that prior knowledge almost certainly plays a role in how an object-colour association is perceived and/or recalled. Speculatively, this may explain why perception of colour for objects (Exp. 1-2, 4-6) was notably poorer than perception of location (Exp. 3). We have added a paragraph to the discussion to elaborate on this point (see quote below; see pages 16, lines 427-435).

"Our analyses assume that participants have no prior associations between the objects and the colours in which they are presented. However, several studies have demonstrated that pre-existing associations do bias how object colour is perceived and recalled [16,74,75]. Speculatively, this may explain why responses for object-colour associations (Exp. 1-2, 4-6) are more prototypical than responses for object-location associations (Exp. 3): we possess substantially more canonical object-colour associations than we do for object-location associations, and this prior knowledge biases colour choices more so than for the equivalent location choices. However, our observation of metacognitive effects for both colour- and location-cued recall suggests that canonical object-colour associations are not a major confounding factor. Nonetheless, future studies may benefit from accounting for canonical associations and including them as covariates to fully decorrelate their influence on behavioural responses."

5. Work by the Kuhl lab on the repulsion of memories with highly similar features (including color hue) appears highly relevant and the paper would benefit from a discussion of how their approaches are related and why they give rise to diverging results.

Thank you for the suggestion. We entirely agree that this is an intriguing discussion point. We feel that the key distinction between our work and that of the Kuhl lab is that their experiments (e.g., Zhao et al., 2021) are designed to maximise competition between stimuli, but the only competition we feel is present in our work is that of prior knowledge (see preceding reviewer comment). Perhaps, when competition is not present, repulsion exerts a smaller effect on retrieval. We have added this to our discussion (see pages 16-17, lines 445-453).

"Here, we have focused on how exemplars are drawn to prototypes as this made up the majority of responses (76.8%; with the remainder being attributed to either repulsion from prototypes or total forgetting). However, other studies have found repulsion to play a larger role in prototype-based distortion (e.g., [18]). We speculate that this is due to differences in experimental design, particularly regarding stimulus competition. For example, when colour is the only dimension in which two exemplars differ, repulsion helps distinguish the stimuli (see [18]; similar competition-driven repulsion can also be observed on a neural level [77,78]). As our experiment did not involve competition between stimuli, we speculate that repulsion has no adaptive benefit here, perhaps explaining why little repulsion was observed."

Minor comments

6. It's difficult to use the figure and figure text alone to understand whether Distance to Target and Distance to Prototype are both using the Recalled Colour to compute the distances – I'd recommending adding this in the figure text to make it very clear.

We have updated the legend of figure 2 to clarify this.

“For the perceptual data, the target distances are computed using the perceptual response and the true, presented colour, while the prototype distances are computed using the perceived response and its associated prototype. For the retrieval data, the target distances are computed using the recalled response and the equivalent perceptual response (disentangling memory biases from perceptual biases; see methods), while the prototype distances are computed using the recalled response and its associated prototype.”

7. The box plots for Retrieval in Figure 3A are shaded darker than all other box plots in that figure e – is this intentional? If so, please explain in the figure text.

We have updated the figure to remove this visual glitch.

Reviewer #3 (Remarks to the Author):

This study investigates the interesting question to what extent we are aware of distortions in episodic memory recall caused by semantic knowledge. Specifically, it remains unclear whether we are metacognitively aware of prototype-based inference when recalling memories. To this end, they developed a novel experimental paradigm in which participants memorize and later recall colour-object pairs. Prototypical colours are identified using K-means clustering, showing that participants to indeed collapse the colour space into “prototypes”. Results over 6 experiments consistently showed that more prototypical responses were associated with decreased confidence, indicating metacognitive awareness of prototypical biases. Several confounds are controlled for such as the location of the response, the type of stimulus (colour and location) and the order of responses and confidence ratings. This is an original study, that is technically sound and reveals interesting and novel findings that will be of interest to memory and metacognition researchers. I only have a few comments.

We would like to thank the reviewer for their positive appraisal and thoughtful comments. In particular, the suggestion of investigating the link between perceptual and recall prototypicality has produced fascinating new results that provide a deeper understanding of how prototype-based distortion impacts memories.

If I understood this correctly, the authors did test whether confidence during encoding predicted later memory bias but they did not investigate whether errors during encoding predicted later bias. If participants already show a prototypical effect for some objects during encoding, are those then also the objects that drive the later memory bias?

Many thanks for this suggestion. It is an important question that we had not considered. We have conducted the proposed analysis and found consistent evidence across all experiments to suggest that the prototypicality of a response during memory formation predicts prototypicality during retrieval. We have added this analysis to the main text in a new section that also contains the results from Experiment 6, where we explored confidence effects during encoding (see quote below; see pages 12-13, lines 286-327).

“Lastly, given that confidence appears to be derived from prototype-based distortions within the memory trace itself, we asked whether the link between confidence and prototype-based distortion may extend to perception/encoding.

To test this idea, we first asked whether there is a relationship between prototype-based distortions observed during perception and those observed during recall. Indeed, we found a positive correlation across participants suggesting that the prototypicality of a perceptual judgment predicts the prototypicality of response for the same stimulus during recall (Exp. 1: $t_{44} = 5.72$, $p < 0.001$, Cohen’s $d = 0.85$, 95% CI = [0.07, 0.14]; Exp. 2: $t_{33} = 4.49$, $p < 0.001$, Cohen’s $d = 0.77$, 95% CI = [0.06, 0.15]; Exp. 3: $t_{27} = 8.10$, $p < 0.001$, Cohen’s $d = 1.53$, 95% CI = [0.09, 0.16]; Exp. 4: $t_{36} = 3.63$, $p = 0.001$, Cohen’s $d = 0.60$, 95% CI = [0.03, 0.11]; Exp. 5: $t_{39} = 3.03$, $p = 0.004$, Cohen’s $d = 0.48$, 95% CI = [0.02, 0.11]). Importantly, as our approach to computing memory-based prototypicality excludes pre-existing perceptual distortions (see methods), this suggests that prototype-based distortions at perception not only persist through to recall but become exacerbated.

We then investigated whether confidence during perception maps onto distortions during recall. In Experiment 6, we asked participants to make a prospective confidence judgement immediately after perceiving/encoding the stimulus, explicitly asking them how confident they felt in their ability to recall the later pairing (“Sure” responses were equivalent to the previous experiments: mean: 62.5%; SEM: 3.6%). Here, a two-factor model continued to reveal a main effect for confidence, where prototypicality was lower for “Sure” responses

[prototypicality for “Sure” responses: 0.51; prototypicality for “Not Sure” responses: 0.53; confidence main effect: $F(1, 36) = 4.46, p = 0.042, \eta p^2 = 0.11$]. While no main effect was observed for epoch [$F(1, 36) = 0.66, p = 0.442, \eta p^2 = 0.02$], we continued to observe the interaction between confidence and epoch, where the difference in prototypicality between confidence ratings was larger during retrieval than during perception [interaction term $F(1, 36) = 22.11, p < 0.001, \eta p^2 = 0.38$; Δ prototypicality at perception (“Sure” > “Not Sure”): 0.02; Δ prototypicality at retrieval (“Sure” > “Not Sure”): -0.05; see figure 4A-B]. Altogether, these results suggest that we possess prospective awareness of the prototype-based distortions that we will encounter during later recall.

Given that both perceptual distortion and prospective confidence correlate with retrieval-related distortion, it is a possibility that prospective confidence does not directly predict memory-based distortion but instead relates to perceptual distortion, which in turn predicts memory-based distortion. To rule out this possibility, we conducted a multiple regression to quantify how well perceptual distortion and confidence can explain memory-based distortion as statistically independent factors. In this model, we found that prospective confidence continued to predict memory-based distortion ($t_{36} = -4.35, p < 0.001$, Cohen’s $d = 0.72, 95\% CI = [-0.08, -0.03]$); though perceptual distortion did not: $t_{36} = 1.56, p = 0.127$, Cohen’s $d = 0.26, 95\% CI = [-0.01, 0.10]$), suggesting the link between prospective confidence and memory-based distortion is not simply an indirect effect relating to perceptual distortion.”

We have also updated the discussion to reflect these changes (see quote below; see page 14, lines 357-367).

“We also reveal that we first become metacognitively aware of prototype-based distortions during memory formation (see Experiment 6), with prospective confidence judgments made at perception correlating with prototype-based distortions at retrieval. This was statistically independent of the correlation between prototypicality links between perception and retrieval, ruling out the possibility of a proxy effect where confidence ratings correlated with perceptual prototypicality, which in turn correlated with recall prototypicality. This complements numerous studies demonstrating that prototype-based distortions first arise during encoding [40–47], and extends these findings to metacognitive awareness. The fact that metacognitive awareness of prototype-based distortions can be observed as early as encoding further questions whether the confidence signal is purely linked to reconstruction-related phenomenon. Rather, it favours the idea that we are metacognitively aware of distortions of the memory trace itself..”

The results from experiment two reveal that samples from a narrower distributions lead to more prototypical responses, during both perception and retrieval. Could this not just be explained by the fact these samples ARE more prototypical, based on the fact that the distances to the prototype are smaller by design? This would also explain why confidence does not seem to have much of an influence in this condition.

We entirely agree with the reviewer. This was our intention when designing the experiment, though feel the phrasing in our original submission may not have been misleading. We have updated the text accordingly (for examples, see the two quotes below; see page 7, lines 133-134; page 11, lines 263-267).

“...as our explicit aim in Experiment 5 was to bias confidence ratings by making colours more prototypical...”

“Experiment 5 manipulated the underlying distribution of the colour samples such that half the samples came from “broad” distributions while the other half came from “narrow” distributions. We reasoned that accurate responses for object-colour associations drawn from the narrow distributions would nonetheless appear prototypical as the absolute distance to the prototype would be small, and this would undermine confidence.”

Minor comments

I found the measure of prototypicality quite hard to understand. It might help to give a few examples of stimuli with high and low prototypicality scores here.

We have included additional descriptions in the methods to help readers comprehend what a high/low prototypical score means (see quote below; see page 6, lines 89-94).

“Prototypicality can be thought of as the extent to which an episodic memory shifts from veridical representation of a stimulus towards a prototypical version. If there is little change in the representation between perception and retrieval, prototypicality is said to be low. If the representation shifts drastically towards the prototype between perception and retrieval, prototypicality is said to be high.”

Line 100: I wouldn’t call a 60/40 split approximately even

We have removed this sentence.

13th May 24

Dear Dr Griffiths,

Your manuscript titled "Metacognitive awareness of memory distortion during recall" has now been seen by our reviewers, whose comments appear below. In light of their advice I am delighted to say that we are happy, in principle, to publish a suitably revised version in *Communications Psychology* under the open access CC BY license (Creative Commons Attribution v4.0 International License).

We therefore invite you to revise your paper one last time to address the remaining concerns of our reviewers and a list of editorial requests. At the same time we ask that you edit your manuscript to comply with our format requirements and to maximise the accessibility and therefore the impact of your work.

EDITORIAL REQUESTS:

SUBMISSION INFORMATION:

OPEN ACCESS:

Communications Psychology is a fully open access journal. Articles are made freely accessible on publication under a CC BY license (Creative Commons Attribution 4.0 International License). This license allows maximum dissemination and re-use of open access materials and is preferred by many research funding bodies.

For further information about article processing charges, open access funding, and advice and support from Nature Research, please visit <https://www.nature.com/commspsychol/article-processing-charges>

At acceptance, you will be provided with instructions for completing this CC BY license on behalf of all authors. This grants us the necessary permissions to publish your paper. Additionally, you will be asked to declare that all required third party permissions have been obtained, and to provide billing information in order to pay the article-processing charge (APC).

* **DATA AVAILABILITY:**

[link redacted]

Best regards,

Jennifer Bellingtier

Jennifer Bellingtier, PhD

Senior Editor

Communications Psychology

Jesse Rissman, PhD

Editorial Board Member

Communications Psychology

orcid.org/0000-0001-8889-5539

REVIEWERS' EXPERTISE:

Reviewer #1: memory specificity and generalization

Reviewer #2: conceptual influences on episodic memory

Reviewer #3: metacognition and perceptual inference

REVIEWERS' COMMENTS:

Reviewer #1 (Remarks to the Author):

The authors have done a nice job of addressing my concerns, and I am happy to endorse for publication. I just note two typographic issues:

pg. 3 line 42: should be 'pre-register' rather than 'pre-registered'.

Table 2: I found it hard to tell how the text lined up across the row.

Reviewer #2 (Remarks to the Author):

The authors have addressed all of my comments nicely. I especially appreciate the extra work to re-analyze the data without repulsions and overshoots, and I think the revised table 2 that was recommended by Reviewer #1 as well as the new Figure panel 3A do a lot to help clarify the manuscript. Looking forward to seeing this published!

Reviewer #3 (Remarks to the Author):

The reviewers have addressed all my previous comments. I especially appreciate the additional analyses and discussion about how the effects might be related to memory encoding/perception.

Reviewer #1 (Remarks to the Author):

pg. 3 line 42: should be 'pre-register' rather than 'pre-registered'.

We have now corrected this (see page 3, line 39).

Table 2: I found it hard to tell how the text lined up across the row.

We have added lines to help clarify which rows each text belongs to (see page 20).